# Interventions to promote the health and well-being of children under 5s experiencing homelessness in high-income countries: a scoping review

Yanxin Tu [1], Kaushik Sarkar,[2] Nadia Svirydzenka,[3] Zoe Palfreyman,[3] Yvonne Karen Parry,[4] Matthew Ankers,[4] Priti Parikh [5] Raghu Raghavan,[6] Monica Lakhanpaul [1]

For numbered affiliations see end of article.

**Correspondence to**
Professor Monica Lakhanpaul;
m.lakhanpaul@ucl.ac.uk

## ABSTRACT

**Objectives** Homelessness among families with children under 5 residing in temporary accommodation is a growing global concern, especially in high-income countries (HICs). Despite significant impacts on health and development, these 'invisible' children often fall through the gaps in policy and services. The study's primary objective is to map the content and delivery methods of culturally sensitive interventions for children under 5 experiencing homelessness in HICs.

**Design** A scoping review guided by the Preferred Reporting Items for Systematic reviews and Meta-Analyses extension for Scoping Reviews checklist.

**Data sources** Databases include PubMed, Medline, SCOPUS, The Cochrane Library and Google Scholar were searched up to 24 March 2022.

**Eligibility criteria** This scoping review includes studies that describe, measure or evaluate intervention strategies aimed at improving child health programmes, specifically those yielding positive outcomes in key areas like feeding, nutrition, care practices and parenting.

**Data extraction and synthesis** Articles were selected and evaluated by two independent reviewers, with a dispute resolution system involving a third reviewer for contested selections. The methodological quality of the studies was assessed using various tools including the Risk of Bias (RoB) tool, Cochrane RoB V.2.0, the Risk of Bias Assessment Tool for Non-randomized Studies (RoBANS) and the Grading of Recommendations Assessment, Development, and Evaluation (GRADE), each selected according to the type of article.

**Results** The database search yielded 951 results. After deduplication, abstract screening and full review, 13 articles met the inclusion criteria. Two predominant categories of intervention delivery methods were identified in this research: group-based interventions (educational sessions) and individual-based interventions (home visits).

**Conclusion** This review highlights effective interventions for promoting the health and well-being of children under 5 experiencing homelessness, including educational sessions and home visits. Research has supported the importance of home visiting to be instrumental in breaking down language, cultural and health literacy barriers.

## STRENGTHS AND LIMITATIONS THIS STUDY

⇒ Adhered to the Arksey and O'Malley framework and the Preferred Reporting Items for Systematic reviews and Meta-Analyses extension for Scoping Reviews checklist, ensuring a focused and methodical approach.

⇒ The employment of diverse methodological quality assessment tools for different study types allows for a thorough and nuanced evaluation of potential biases, increasing the reliability of this review's conclusions.

⇒ Unlike systematic reviews, this review did not conduct formal data synthesis, potentially limiting the comprehensiveness of evidence overview.

⇒ High heterogeneity, resulting from the inclusion of various study designs, and low generalisability due to the restriction of studies to those conducted in HICs, limits the comparative and broader applicability of the results.

## INTRODUCTIONS

As per the McKinney-Vento definition of homeless, homeless children and youths indicates individuals who lack a fixed, regular and adequate night-time residence.[1] This definition includes children and youths who are doubling up with others due to housing loss or economic challenges, residing in motels, hotels, trailer parks, or camping grounds due to lack of alternative adequate accommodations, living in emergency or transitional shelters, or abandoned in hospitals. It also covers those living in places not typically used for regular sleeping, such as cars, parks, public spaces, abandoned buildings, substandard housing, bus or train stations, as well as migratory children living in similar conditions.[2] The UN Office of the High Commissioner for Human Rights acknowledges that homelessness has 'emerged as a worldwide human rights concern', especially in high-income

nation-states with the means to address it.[3] Homelessness among families with children is now a growing problem in high-income countries (HICs). Countries with Gross National Income per capita of US$13 846 or more are defined as HICs.[4] Between 2014 and 2018, family homelessness almost doubled in Ireland, rising from 407 to over 1600 families. Between 2006 and 2013, New Zealand had a 44% rise in family homelessness. In 2019, the USA had around 54 000 families with children, accounting for one-third of the country's homeless population.[5] In 2019, the charity Shelter reported that a child loses their home every 8 min in Great Britain, which is the equivalent of 183 children per day.[6] According to the Children's Commissioner, there might be as many as 210 000 homeless children living in temporary accommodation, or couch surfing in England, as well as roughly 585 000 people who are either homeless or in danger of becoming homeless.[6]

In particular, the plight of children aged 5 years or less residing in temporary accommodation is often overlooked or under-recognized. There is a lack of policy supporting them since they are not seen on the streets as homeless. However, many of the children have pre-existing conditions such as epilepsy, asthma, anxiety and diabetes, and potentially are the most susceptible to viral infection.[6] Moreover, the first 5 years of life is a critical duration for the optimal development of the brain. This is especially important for children who experience poverty/housing/transient lifestyle, as this places them at risk of failing to reach the full development of the brain, potentially leading to many health concerns and issues with language development and motor skills and social problems.[7]

Our partnership initially concentrated on evidence-based and community-based participatory engagement approaches that enhance the health and well-being of children under 5s who are or at risk of being homeless. This was informed by an investigation into how services could adapt to, and learn from, global public health interventions and family experiences, the purpose of which was to first tailor current health strategies for effective outreach to populations who are homeless or at risk of homelessness. Then, second, in light of both existing and new findings, provide easily accessible resources to health and social care professionals, as well as families of 'invisible' children. These children were excluded from research and national policies, as they are often not counted by services, or are not seen as existing as 'homeless'. However, current literature did not elucidate on the existing intervention strategies, the delivery channels used by these strategies, the language and cultural barriers encountered, the methods for circumventing these barriers, and the creation of acceptable materials for children under 5s. To guide the literature search and review, the authors predefined several themes of importance for the health and well-being of homeless children: feeding and nutrition, care practices, dental care, mental health and well-being, safe sleeping, physical activity and parenting support. These themes, identified as common problems in homeless populations and known to have been effectively addressed in other low-income and middle-income countries,[8 9] served as the foundation for this scoping review. These themes are critical for the integrated health of children under 5s living in temporary accommodation, and there is a complex need to incorporate all these aspects to address or alleviate the current situation of homelessness.

The primary objective of this review is to map the content, and method of delivery of interventions that are culturally sensitive and accessible for populations at the crossroads of poverty/housing/transient lifestyle and focused on children under 5s. Specifically, our study aims to address the following points:

▶ Develop inclusive and engaging practice interventions for populations living in temporary accommodation/homelessness, with children under 5s.
▶ Identify the potential critical points of contact. These refer to individuals who play pivotal roles in delivering health services or interventions and engaging with the target population. They serve as key intermediaries, bridging the gap between the health system or intervention and the individuals it aims to benefit.
▶ Understand how to communicate with mobile populations who have poor health knowledge, literacy, and/or language barriers.
▶ Identify methods of creating appropriate, acceptable and accessible communication materials.

## METHODS

This focused and methodical scoping review, which was enriched by insights and real-world experiences from global experts, followed the guidelines set forth by the Arksey and O'Malley framework[10] as well as the Preferred Reporting Items for Systematic reviews and Meta-Analyses (PRISMA) extension for Scoping Reviews checklist for scoping reviews.[11] This review follows a predefined protocol (see online supplemental material 1). There was no patient and public involvement in this study. In aligning with the objectives and methodology of a scoping review, this study embraces an exploratory approach, encompassing a wide range of topics related to the health of homeless children.

### Eligibility criteria

The eligibility criteria for our scoping review are structured to include studies that are in English, published between 2000 to 2022, and conducted in HICs as defined by the World Bank.[4] The study focused on interventions targeted at children aged 5 years or less from marginalised or socially excluded families/population groups. It included studies that described, measured or evaluated a pilot or implementation of a strategy, tactic, process and/or method targeted at improving child health programmes, with a specific focus on mapping the culturally sensitive approaches. The interventions included were those that demonstrated improvements in outcomes such as service coverage and optimisation, access, utilisation

and specifically in the nine identified areas of (1) feeding, (2) nutrition, (3) care practice, (4) parenting, (5) dental, (6) well-being, (7) mental health, (8) safe sleeping and (9) physical activity. Studies in languages other than English, from non-HICs as per the World Bank, targeting children over 5 years, involving non-marginalised groups, or those not demonstrating improvement in the outlined nine areas were excluded. By adhering to these eligibility criteria, the scoping review will ensure that the findings are based on the highest quality evidence and are relevant to the topic of interest. In case the number of these studies is less than 30, we will broaden our investigation to include studies on interventions aimed at children over the age of 5, but only if they offer valuable insights or strategies that can be adapted for younger age groups.

### Information sources and search strategy

The databases searched from inception of this project up to 24 March 2022 included: PubMed, Medline, SCOPUS, The Cochrane Library (Cochrane Database of Systematic Reviews, Cochrane Central Register of Controlled Trials (CENTRAL)), and Google Scholar. One researcher (YT) developed the search strategy after preliminary deliberations and consensus within the review team. Conference abstracts and Third Sector Reports were also searched for grey literature. Experts from Donor/Philanthropic organisations and other key experts were contacted for any additional published or unpublished work.

The search strategy was developed to ensure that any relevant studies were identified helping to make the review exhaustive. In the search for literature on the homeless population, two concepts were used: 'marginalisation' and 'social exclusion'. The search strategy was comprised of key search terms drawn from current search strings and customised for each electronic database. We also conducted the relational and citation search to screen cited articles of the first iteration of selections, followed by citations and related articles for each inclusion in PubMed, Google Scholar and PubMed, respectively.

### Study selection

Search results were uploaded to the Covidence software management system[12] where several reviewers screened content. All articles were extracted and compiled in a single spreadsheet. The spreadsheet was equipped with duplicate study filters, and inclusion filters (inclusion criteria not managed through Search Term) and selection columns. Inclusion and selection filters were filled in independently by one reviewer (YT). All studies selected by two reviewers in Covidence were included, while all studies rejected by two reviewers in Covidence, were rejected. All studies marked for inclusion by one reviewer were marked as disputed selection. In the case of disputes, a third expert reviewer (ML) made the final inclusion decision. Two round of screening was undertaken: title-abstract (TiAb) and full text (FT). The reasons for exclusion was recorded and mapped in a flow chart, as per PRISMA guidelines.

### Data charting process

A data extraction form (DEF) was developed as per CEB Critically Appraised Topics (CAT) guidelines by KS and piloted by YT on the first six article included. The piloting essentially informed whether the DEF could extract necessary and sufficient information as per the objectives set. The form was then amended based on the pilot and the finalised form used to extract data from full-text articles. The authors abstracted data on article characteristics (target geography, study design), demographic characteristics (age and gender) and intervention characteristics (number of participants, number of sites, type of setting, duration, channels and agency of delivery, service point, end-users).

### Methodological quality appraisal

The methodological quality of included articles was assessed using different tools, relative to the article type. The assessment of Risk of Bias (RoB) tool was only used for interventional studies when results were quantitative analysis or pooling. GRADE[13] was used for assessment of bias in selective publications and selective non-reporting for systematic reviews. For randomised trials, Cochrane RoB V.2.0[14] was used to assess multiple sources of bias and RoBANS[15] was used for assess bias due to selective non-reporting and bias in the selection of the reported result. All studies were critically appraised as per the CAT grading.

### Data synthesis

The data from the included studies were collated, and a summary table was constructed to present the characteristics and conclusions of each study (table 1). This table encapsulated key information including study authors, age group targeted, geographical location, sample size, contact points, intervention outcomes and their primary focus. The integration of our findings focused on extracting meaningful insights from the commonalities and variances across different studies. This included identifying effective communication strategies and contact points that are crucial in reaching and engaging families and children in need. We synthesised the information to identify common themes, participants characteristics, delivery methods, critical point of contact and communication strategy with poor health literacy population.

### Patient and public involvement

No patient involved.

## RESULTS
### The literature search

The database search yielded a total of 951 results; no additional articles were found in the grey literature search. 523 records were evaluated after deduplication. After screening title and abstract, 433 relevant articles were excluded. This resulted in 90 articles for review, of which

**Table 1** Summarisation in data extraction of included studies

| Authors | Age | Location | Sample size | Contact points | Intervention focus |
|---|---|---|---|---|---|
| Burgi et al[28] 2012 | 5.2 years (SD=0.6) | Switzerland | n=652 | Health promoters (volunteers) and preschool teacher | Sessions of physical activity (PA) for children, workshops for teachers, interactive information and discussion evenings for parents |
| Foka et al[25] 2021 | 7–12 years | Greece | n=72 | Trained facilitators (volunteers) | Resilience-building programme with group-focused interactive educational activities |
| Goodman et al[18] 2022 | 6 months | USA | n=1243 | Trained nurses and staff | A home visit of 2 hours duration is implemented with the aim to enhance the health and overall well-being of families having infants and young children |
| Grace et al[16] 2019 | 15 months | Australia | n=363 | Trained volunteers | Weekly 2-hour home visits by trained volunteers, targeting social isolation, community connectedness and parenting skills |
| Gross et al[17] 2009 | 2–4 years | USA | n=292 | Trained volunteers | 11 weekly video scenarios and group discussions for parental behaviour enhancement |
| Guerrero et al[20] 2021 | 0–12 years old | USA | n=1749 | Trained staff (volunteers) | Group-based educational sessions on stress, depression and parenting |
| Holtrop and Holcomb[21] 2018 | 3.18 years old (SD=1.66) | USA | n=12 | Marriage and family therapist | Parent management training encourages parenting strategies through educational sessions, practice tasks, and satisfaction surveys |
| Melley et al[22] 2010 | 2 weeks–3 years old | USA | n=87 | Trained nurses and parents | Enhanced children's resilience in homelessness through interactive therapy sessions |
| Ristkari et al[23] 2019 | 4 years old | Finland | n=463 | Professionals in healthcare and social services | Web-based parent training for basic and practical positive parenting skills, with follow-up telephone coaching to improve behaviour |
| Rowe et al[24] 2012 | 0–4 years old | Australia | n=116 | Early childhood professionals and trained nurses | Tweddle Child & Family Health Service (TCFHS) and Day Stay Program (DSP) offer individual, and group level educational sessions for positive parenting skills, focusing on sleep, feeding and establishing routines |
| Dugravier et al[19] 2013 | prenatal to 24 months old | France | n=440 | Phycologists | Psychologists' home visits for mental wellness, attachment and depression management over 14 sessions |
| Spijkers et al[59] 2010 | 9–11 years | Netherland | n=160 | Trained nurses | Multilevel parenting programme promoting children's social, emotional, and behavioural development through consultations, educational sessions, home visits and parent–child interactions. |
| Yousey et al[27] 2007 | 18 months–6 years old | USA | n=56 | Trained nurses | Improving homeless children's nutrition by educating parents and shelter food providers through sessions developed by dietitians |

77 were excluded, as they did not meet the inclusion and exclusion criteria (figure 1).

### Data summary and synthesis

Data from included studies were collated, and table 1 presents characteristics and conclusions reported.

### Quality review

Based on the screening results of Cochrane RoB V.2.0, the study by Grace et al exhibited a risk of performance bias. Caregivers and personnel administering the interventions were aware of participants' allocated intervention throughout the trial, as this information was disclosed by participants in their questionnaire responses about their experiences with services.[16] Furthermore, there was a risk of selection bias in the study by Gross et al due to issues with the randomization process. Parents were not randomly assigned to different intervention levels, but instead chose how many group sessions they attended. Consequently, observed improvements in child behaviour might be attributed to factors associated with parental attendance, rather than the intervention itself.[17] Study of Goodman et al is limited by the use of self-report data, which could contribute to recall and social-desirability bias.[18] Concerns were also raised regarding bias due to missing data for study by Dugravie et al. Only half of the participants completed their perinatal home-visiting programme.[19]

In light of the screening results of RoBANS, study of Guerrero et al is also constrained by the use of self-report data.[20] The challenge of retaining parents was

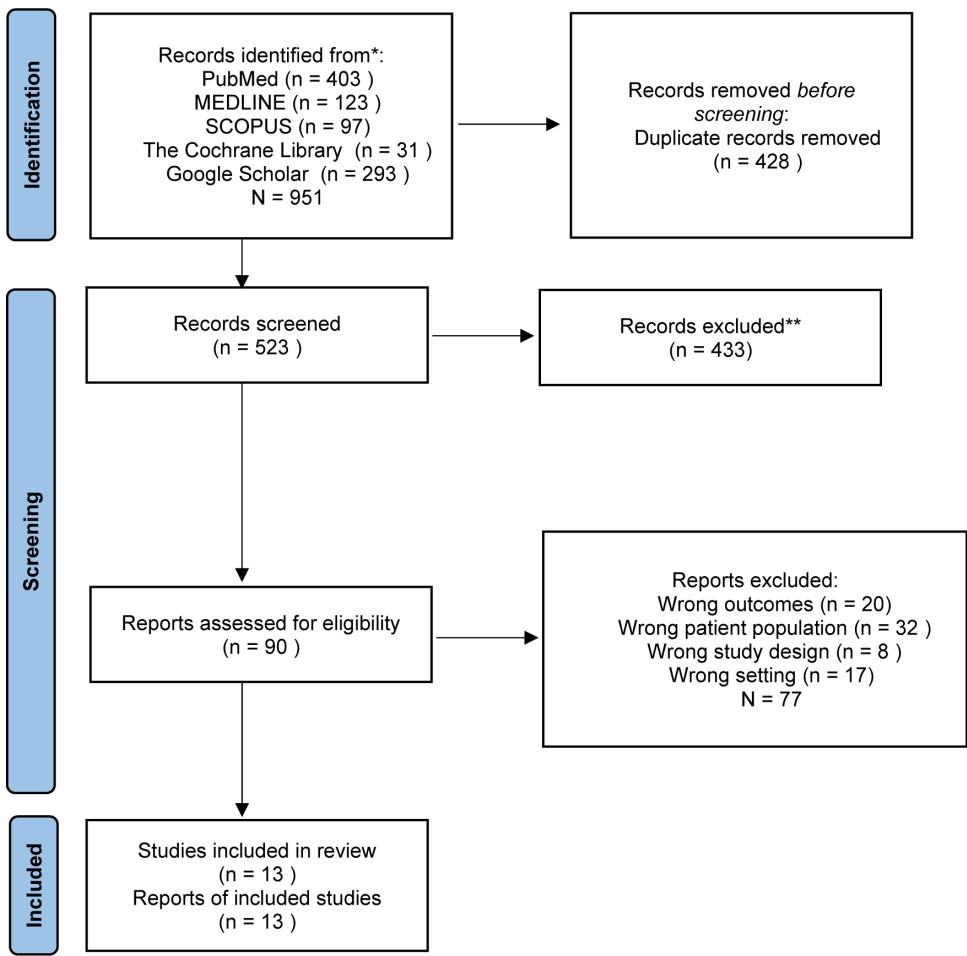

**Figure 1** Preferred Reporting Items for Systematic Reviews and Meta-Analyses (PRISMA) 2020 flow diagram.

also a notable drawback of the research by Holtrop *et al*, since only 50 of the participants attended more than fifty percent of the sessions.[21] In general, we did not exclude any studies based on risk-of-bias assessment.

### Characteristics of intervention programmes
#### Intervention themes
In total, 13 studies were included for review (table 1). Of the nine targeted intervention themes in the inclusion criteria, none pertaining to dental outcomes for marginalised children in HICs were identified in the search. Several overlapping themes, such as feeding, nutrition, safe sleeping and mental health, were commonly addressed through enhancing positive parenting skills and support. Overall, four intervention theme categories were identified from the evidence synthesis.
► Positive parenting skills (sleeping and settling, safe sleep, feeding and meal time, establishing routines, attachment and boding).[17 18 21–24]
► Mental health and well-being (social isolation, chronic stress and depression, toddler behaviour).[16 19 20 25 26]
► Nutrition (improve nutrition knowledge of parents).[27]
► Physical activity (life style, adiposity and fitness).[28]

### Characteristics of participants
On an initial exploration of the literature, the number of studies within this age range was limited. Given the importance of our research question and the potential benefits of gaining a broader understanding of the available interventions, this study encompassed studies involving children above five. The majority of the studies encompassing children aged under 5-year-olds,[16–24 27 28] 2 studies included were targeted on early childhood (ages 6–11).[25 26] The justification for this approach lies in the potential transferability of interventions for slightly older children to younger age groups.

Gender distribution is generally even and homogeneous across different study. However, in a nonrandomised intervention study by Holtrop *et al*,[21] the sample consisted of 75% females and 25% males. Gender distribution was not specified in the studies by Yousey *et al* and Guerrero *et al*.[20 27] The backgrounds of the participants varied widely across the included studies. Participants encompassed children of migrant and/or parents with low education levels (ELs), children from homeless or poverty-stricken families, newborn infants from transitional housing communities and families living in low socioeconomic areas.

## The method of delivery

Detailed description of the intervention programmes are shown in table 1. The intervention duration varies from 1 week to 2 years, with four included studies having intervention durations of over 12 months,[16 17 19 28] four studies ranging from 6 to 12 months,[18 20 21 26] and five studies lasting less than 6 months.[22–25 27] The interventions identified in this study have been broadly categorised as follows:

► Education-based group interventions, performed as group-based intervention, such as those based on the Webster-Stratton Model[29] and the Oregon Model,[30] are effective in improving parenting practices and child behaviour, especially in socioeconomically disadvantaged communities.[17 21 23 24 27 28] These include methods such as web coaching, videotaped vignettes, group discussions and family meetings. Broadening the intervention scope to include other involved adults and offering education on nutrition, physical activity, media use and sleep further boosts their efficacy. Programmes like the Strengths for the Journey in refugee camps have used positive psychology concepts,[25] while the Parent–Child Interaction Therapy model[31] successfully strengthened parent–child relationships in shelters. Overall, these education-based group interventions offer a cost-effective, holistic approach to child welfare.

► Home-visiting interventions involved trained professionals offering support and education to families with young children in their homes, with interventions being adapted to each family's unique needs, often complemented by group-based interventions. The goal was to foster community connectivity and independent utilisation of services.[15 18 25] Some studies, such as Guerrero *et al*'s work, supplement these home visits with group-based education, training staff and parents in stress and depression management via a flexible, Train-the-Trainer approach.[16] Additionally, the Positive Parenting Program employs a multi-level system of family intervention combining home-visiting and education session approaches, escalating in intensity based on the severity of the child's behavioural and emotional issues.[26]

## Critical points of contact

Marginalised children may face a variety of barriers that can prevent them from accessing healthcare, education, social services and other resources that are necessary for their well-being. Designing contextual, culturally sensitive and diverse interventions remains a challenge. However, identifying effective points of contact for interventions can help overcome these obstacles. Studies have used maternal and child health nurses,[18 22 24 26 27] and health professionals with relevant expertise,[19 21 23 24] as critical points of contact. Trained volunteers, often acting as health promoters or facilitators, have also served this role.[16 17 20 25 28] Nurses and trained volunteers are cost-effective, easy to train and access. Health professionals typically possess greater knowledge, skills and experience, adhering to strict professional standards. It was hypothesised that professionals who were more highly trained in interested field would be more competent in implementing interventions precisely and effectively.[11] The choice of critical points of contact hinges on the study's complexity and scope, warranting a cost-effectiveness analysis.

## Communication with poor health literacy population

This review notably aimed to explore strategies for creating suitable, acceptable and accessible communication materials, especially for mobile populations with limited health knowledge, literacy, or language skills, particularly those with children under 5. In relation to the development of such materials, the Chicago Parent Program, as described in the study by Gross *et al*, devised an intervention content and strategy. Communication materials were created in partnership with an advisory council of parents, including seven African-American and five Latino parents from different neighbourhoods in Chicago. This council provided input to the programme creators about their parenting challenges, the scenarios they would prefer to see in video format, and the best ways to present parenting techniques that align with their values, lifestyle and cultural norms. This approach made the content patient-oriented, readily acceptable and culturally sensitive.[17] Yousey *et al* and Guerrero *et al* discussed making teaching materials, handouts and class activities in a low-literacy format, visually appealing (with pictures and varied colours), and game-oriented (puzzles, riddles) as useful strategy that make acceptable, and accessible communication material.[20 27] As materials developed for marginalised parents with poor health literacy or low ELs, it is important to provide information in different languages. Burgi *et al* also equipped native speakers of the main foreign languages to answer any questions from parents throughout the intervention to break language barriers.[28] In terms of the communication method with marginalised population, the reviewed articles emphasised the importance of home-visiting, as a less formal, relationship-based approach is potentially crucial in overcoming barriers to service engagement, such as language and cultural obstacles, and limited health literacy.[16 18 19 26]

## DISCUSSIONS
## Group-based interventions

Educational sessions were the most commonly employed strategy in group-based interventions included in this scoping review. Providing educational sessions in group level can be a valuable and effective approach to supporting families with children under 5, who experience homelessness in HICs. Families experiencing homelessness may feel isolated and disconnected, and participating in a group-based intervention can provide an opportunity for them to connect with other families in similar situations.[16] Therefore, group-based interventions can provide families with a sense of social support and

community. Educational sessions can also be tailored to meet the specific needs of families experiencing homelessness with children under 5, since the content of sessions are dependent on the health needs. In short, group-based educational session interventions can empower families experiencing homelessness by providing them with the tools, knowledge, and support they need to improve their situation.

Web-based sessions are an innovated form of carrier of education and thus modifying behaviours and improve health outcomes in children under 5 experiencing homlessness. Parent training programmes that use technology can have numerous advantages compared with conventional interventions, including improved consistency, enhanced availability, increased convenience and a reduction in both time and financial expenditure.[32][33] The concept of technology-based parent training is not new; dating back to 1988, Webster-Stratton and his team[29] used video recordings as the main medium to conduct a parent training initiative. The innovation of web-based sessions offers an efficient platform for imparting necessary knowledge, modifying behaviours and improving health outcomes in the children under 5 living in temporary accommodation. Given the high usage of cell phones among the homeless population in HICs, the potential for web-based interventions becomes even more evident. This suggests a promising intersection between technology and accessible parent training programmes within the context of homelessness. Evidence indicated that most people experiencing homelessness have cell phones in HICs. In a case report conducted in Los Angeles, USA, 85% of homeless people used a cell phone and used text messaging daily, and 51% accessed the Internet on their cell phone.[34] A number of other studies have postulated a convergence on the similar findings.[35–37] The current cell phone using status support the possibility of Web-based intervention use in the homelessness context. Web-based interventions can potentially eliminate the obstacles related to in-person sessions, allowing individuals to pursue assistance for mental health issues without the concern of stigma.[38–40] The web-based intervention was demonstrated to be low cost, low threshold and of great implications for evidence-based interventions in the future. Accordingly, recent studies in the sphere of parent training have pivoted towards online training programmes. These interactive online training platforms for parents can successfully surmount numerous obstacles typically encountered during the execution phase of traditional programmes, suggesting a promising intersection between technology and accessible parent training programmes within the context of homelessness.[23]

### Individualised interventions

Parents, and particularly mothers, are susceptible to social isolation, especially during the initial transition to parenthood when they may experience intense feelings of fatigue or a lack of readiness. In research concerning parents facing additional challenges, such as parenting a child with a disability, recent immigration, experiencing cognitive or mental health difficulties, the prevalence of social isolation was found to be significant.[41–43] To address this issue, one commonly used strategy is home visiting. Typically coordinated by a local organisation, this intervention involves assigning a volunteer or professional to provide social support to individuals identified as needing additional help. Through regular visits, these health providers offer a variety of support services and work towards enhancing the individual's engagement with formal services, fostering greater community connection and promoting independent utilisation of services.

Individualised interventions (home-visiting) can be more flexible than group-based interventions, as they can be adapted to meet the changing needs of families over time, provide personalised support tailoring to different family's needs, provide families with access to a range of resources and services, and provide a greater sense of confidentiality and privacy. These individualised approaches are instrumental in dismantling barriers to service engagement, including those tied to language and culture.[44] Previous studies advocate for the possible benefits of home visits by non-professionals, such as facilitating the dissemination of health-related information,[45] enhancing social networks for those in isolation, fostering emotional well-being and parental proficiency, and endorsing positive health outcomes.[46] Furthermore, previous research has emphasised the value of home-visiting, a less formal, relationship-based approach that complements other more structured services in the service landscape.[44] This idea suggests that combining group-level educational sessions with home visits could reinforce training content by allowing for individual feedback collection and catering to personal needs. Given these potential benefits of individualised interventions, the strategy of home-visiting has gained attention in both research and practice. By bridging the benefits of both individualised interventions and community support, home-visiting appears to be a promising strategy that could be more widely applied in future intervention designs for homeless population.

### Research gap

While there's a growing focus on understanding barriers to dental services for the homeless population in HICs, it is evident that service provision, particularly for homeless children, is woefully inadequate.[47–49] Studies have shown a lack of dental healthcare services for the homeless across governmental, private and third-sector levels.[47] Moreover, current peer-reviewed literature is sparse on strategies improving access to, and uptake of, dental care for this marginalised group.[50] A scoping review of grey literature identified only two services specifically catering to the homeless population in Australia.[50] The effectiveness of these services remains unknown, raising questions about their generalisability across various geographical settings or age groups. The issue is pressing considering the vulnerability of marginalised children to adverse

dental problems. An estimated three million children in Europe are believed to receive inadequate dental treatment,[51] and children from low-income families are twice as likely to have cavities compared with those from higher-income households.[52] Consequently, there is an urgent public health need to implement targeted dental services for young children within the homeless population.

In the context of our scoping review, another noticeable gap is that only one study focused specifically on the theme of physical activity.[28] This paucity of research is a concern given the global public health issue of childhood overweight and obesity.[53] In addition, children of migrant parents and those with low socioeconomic status are considered high-risk groups for the development of obesity and of low fitness, which is known as the hunger–obesity paradox.[54] In addressing this issue, it was scoped that Burgi *et al* developed a lifestyle intervention in Switzerland aimed at enhancing physical activity level in preschool children, primarily from migrant or lower EL families.[28] This intervention showed beneficial effects, yet it remains an outlier; overall, interventions in these populations have shown less effectiveness.[55–58] Given the magnitude of the obesity and low fitness issue among homeless populations, the paucity of targeted interventions underlines a pressing need for further research. Future studies should aim to develop effective, evidence-based approaches that integrate health promotion programmes within the broader context of social and cultural values. This could help in designing interventions tailored to this vulnerable population, thereby addressing this significant gap in the research.

## Strengths and limitations

This study represents the first scoping review of peer-reviewed literature dedicated specifically to intervention approaches for marginalised children living in temporary accommodation under 5 years old across various themes. We deployed a rigorous strategy to extensively map available evidence, with the aim of identifying knowledge gaps to inform future research. The quality of all included studies was high, underscoring the credibility of our findings. Despite these strengths, several limitations should be acknowledged. Unlike systematic reviews, scoping reviews typically do not conduct formal data synthesis. Consequently, the results may not provide as comprehensive or robust an overview of the evidence as systematic reviews do. Furthermore, this review exhibits a high level of heterogeneity due to the incorporation of studies with diverse designs and methodologies, which can make comparing and synthesising results challenging. Lastly, this review's generalisability may be limited due to the typically small scale of interventions in the included studies. This focus on HICs could also limit the generalisability of the findings to countries with different socioeconomic contexts and resource availability.

## CONCLUSIONS

This review sheds light on health interventions that effectively reach children under 5 years old who are homeless or at risk of homelessness. It contributes not only to the literature but also provides actionable resources for health and social care professionals and the families of these often 'invisible' children. While a robust body of research focuses on parenting support, mental health, well-being, nutrition and feeding, and care practices, we found a significant gap in addressing dental health and overweight within marginalised families in HICs. The review of 13 interventions revealed that 2 primary methods, group-based educational sessions and individual-based home visits, hold promise. Notably, an integrated approach combining these two methods appears to be feasible. It was also identified that health professionals, nurses and trained volunteers play crucial roles in delivering these interventions. Despite these insights, several areas necessitate further investigation. For example, a comparison of cost-effectiveness between group-based and individualised interventions remains elusive. Moreover, the review identified a common issue of low community connectedness and service utilisation, highlighting the need to explore ways to enhance community resource usage in HICs further.

**Author affiliations**
[1]Population, Policy and Practice Research and Teaching Department, University College London Great Ormond Street Institute of Child Health, London, UK
[2]Aceso Global Health Consultants Limited, London, UK
[3]School of Applied Social Sciences, De Montfort University Faculty of Health and Life Sciences, Leicester, UK
[4]College of Nursing and Health Sciences, Flinders University, Adelaide, South Australia, Australia
[5]The Bartlett School of Sustainable Construction, University College London, London, UK
[6]School of Nursing and Midwifery, De Montfort University Faculty of Health and Life Sciences, Leicester, UK

**Contributors** Each author made substantial contributions to the design and execution of the study, and to the writing and revising of this article. ML, as the guarantor of this work, accepts full responsibility for the work and the conduct of the study, has access to the data, and controls the decision to publish. YT played a primary role in conceptualising the study, data collection and analysis, and drafting and revising the manuscript. KS, ML and NS guided the study protocol and developed the search strategy. KS was actively involved in data collection and analysis. ML and NS were key contributors to the study conception and design, guided data analysis, and revised the manuscript. PP, ZP, YKP, MA and RR were responsible for all major areas of concept development and study planning, were consulted on the data analysis and its interpretation, and provided manuscript edits. All authors reviewed and finalised the manuscript.

**Funding** This work was supported by the ESRC as part of UK Research & Innovation's rapid response to COVID-19 (ES/V016253/1).

**Competing interests** None declared.

**Patient and public involvement** Patients and/or the public were not involved in the design, or conduct, or reporting, or dissemination plans of this research.

**Patient consent for publication** Not applicable.

**Ethics approval** Not applicable.

**Provenance and peer review** Not commissioned; externally peer reviewed.

**ORCID iDs**
Yanxin Tu http://orcid.org/0009-0003-8426-7019
Priti Parikh http://orcid.org/0000-0002-1086-4190
Monica Lakhanpaul http://orcid.org/0000-0002-9855-2043

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
