## [Reviewer comments · BMJ Open]

ARTICLE DETAILS

TITLE (PROVISIONAL)	Interventions to promote the health and well-being of children under 5s experiencing homelessness in high-income countries: A Scoping Review
AUTHORS	Tu, Yanxin; Sarkar, Kaushik; Svirydzenka, Nadia; Palfreyman, Zoe; Parry, Yvonne; Ankers, Matthew; Parikh, Priti; Raghavan, Raghu; Lakhanpaul, Monica

VERSION 1 – REVIEW

REVIEWER	Katrina Milaney University of Calgary Cumming School of Medicine, Department of Community Health Sciences
REVIEW RETURNED	03-Oct-2023

GENERAL COMMENTS	A trainee named Angelina Adams assisted with this review. Summary This is a scoping review which included data from 13 studies. The objective of the review is to map the delivery methods that are accessible and available for populations in high income countries, who struggle with homelessness and is mainly focused on children 5 years of age and under. There were 6 studies from North America, 5 studies from Europe and 2 studies from Australia making this review generalized in population and location. The findings suggest that identifying the need for more enhancement of community resources in high income countries is critical while there is an importance of house interventions and community based educational interventions. Overall comments This is an important topic of study overall, I think that the manuscript is well done, with some minor improvements to strengthen the comprehension of the study. I would suggest that the authors clarify what their study design is. Both scoping review and systematic review language was used interchangeable within the manuscript. More clarification can outlaw the measures from the study design that is not applicable to this manuscript. The introduction provided a great synopsis and understanding of the study; however, I think the introduction could be improved by adding in more information about what is homelessness and how do they measure homelessness in this study especially for there study population of 5 years and younger. The study interchangeable talks about children with pre-existing health conditions, children who come from poverty, lack of housing and little to no support for health development. With more clarification of their understanding and definition of homelessness criteria this would improve the accuracy
---

	and reliability of the study. What makes this study engaging and interesting is that its objective is to highlight interventions for more health promotion as there is concern for children experiencing homelessness globally. The population of the study is unclear as there is a generalization in high-income countries, and as there are many variables that make these high-income countries different from one another, with more clarification and description this would enhance the study. The authors outlined the inclusion and exclusion criteria that was important for their literature search, I thought that it would be crucial to include the theme culture in the methods section of the eligibility criteria. As stated in the objective that this study it is to highlight culturally sensitive health promotion interventions for children. With no criteria or literature specifically including culture or culturally interventions it is troubling for the methods and results sections. Within the discussion there are 3 articles cited that are older literature. Line 29, page 17 of the manuscript the author explained “recent” literature with finding and cited the article from 1993 [43]. I would recommend that the author comments about the literature and how it is foundational to be apart of the manuscript, and what about it this article makes the findings recent. I would recommend that the author explain the 2 other articles used in the discussions that are older literature or emerge in using other articles
--	---

VERSION 1 – AUTHOR RESPONSE

-Homelessness Definition and Measurement: We have added more clarification on the definition of homelessness as described in the McKinney-Vento Act to the introduction section. This includes specific measures of homelessness for our study population of children aged 5 years and younger.

-Clarification of Study Design: We have clarified further that this study is a scoping review guided by the PRISMA-ScR checklist. Terms that might have been confusing and suggested a systematic review approach have been removed to prevent any misunderstanding between a scoping review and a systematic review.

-Inclusion of Culture Theme in Methods: In Table 1 (Summary of Eligibility Criteria for Literature Search), we have now included a focus on culturally sensitive approaches, aligning with our objective to highlight culturally sensitive health promotion interventions for children.

-Contextualizing Article 43: Article 43 is indeed not a recent study. We have revised the description of Article 43 from 'recent' to 'previous'. This article, although not recent, provides historical context and serves as a foundational study supporting the role of non-professionals in community-based interventions on child health. The reason for citing this study is its use of a randomized controlled trial design to evaluate the effectiveness of non-professional volunteers in delivering child development programs. This methodological rigor was groundbreaking at the time and provides a reliable basis for comparison with more recent studies. It is an early foundational study that supports the role of non-professionals in community-based interventions on child health.

-Clarification on Study Population: We acknowledge the generalization in high-income countries (HICs) and have added further clarification in the strengths and limitations section. We recognized that mapping intervention strategies only for HICs could be limiting and have clarified this in the revised manuscript. However, by concentrating on HICs, we were able to explore deeply into contexts where healthcare systems and social support structures are relatively well-developed, thus providing

a clear benchmark for effective interventions. This is still very valuable in informing other HICs about effective intervention strategies for homeless children.

We trust that these revisions comprehensively address your concerns and enhance the manuscript's clarity and robustness. Thank you for the opportunity to improve our work.

VERSION 2 – REVIEW

REVIEWER	Katrina Milaney University of Calgary Cumming School of Medicine, Department of Community Health Sciences
REVIEW RETURNED	12-Dec-2023

GENERAL COMMENTS	Summary This scoping review included data from 13 articles that met the inclusion criteria. The objective of the review is to map the culturally sensitive programs for populations in high income countries, who struggle with homelessness and are children 5 years of age and under. There were 6 studies from North America, 5 studies from Europe and 2 studies from Australia making this review very broad in terms of generalization in population and location. The findings highlight that identifying the need for more enhancement of community resources in high income countries is critical while there is an importance of home interventions by breaking down the barriers in language, culture and health literacy. Overall Comments This is an important topic of study overall, I think that the manuscript is well done, the minor improvements to strengthen the comprehension have been changed, creating a nice flow to the article. The authors clarified in the limitations that the article was not a systematic review because it did not conduct a formal data synthesis however stated this could limit the comprehensiveness which has provided greater clarity and reflection to this article. The introduction was improved significantly by adding more information on the definition of homelessness, and the use of measurement for the purpose of this study. This has improved the accuracy and reliability of the study by understanding the study population in a clearer sense for inclusion. In the discussion the authors added previous studies versus recent research which helps the reader understand the foundational part of using those studies within the article. I believe in the revision the authors did a great job at stating the significance of the studies used. At the end of the article the authors discussed the limitation of generalizability. I believe this was a great concept to point out because it can be troublesome only looking at high income countries and comparing it globally. The edits that have been made to this study have created more flow and reliability. I don't think there needs to be any more revisions made to the study.
---